# Breast, Prostate, Colorectal, and Lung Cancer Incidence and Risk Factors in Women Who Have Sex with Women and Men Who Have Sex with Men: A Cross-Sectional and Longitudinal Analysis Using UK Biobank

**DOI:** 10.3390/cancers15072031

**Published:** 2023-03-29

**Authors:** Sarah Underwood, Georgios Lyratzopoulos, Catherine L. Saunders

**Affiliations:** 1Primary Care Unit, Department of Public Health and Primary Care, University of Cambridge, Cambridge CB2 0SR, UK; 2Epidemiology of Cancer Healthcare and Outcomes (ECHO) Group, Department of Behavioural Science and Health, University College London (UCL), London WC1E 7HB, UK

**Keywords:** cancer epidemiology, cancer risk, cancer incidence, inequalities, sexual minority health

## Abstract

**Simple Summary:**

In this analysis, we look at the number of new breast, lung, colorectal and prostate cancer diagnoses among women who have sex with women and men who have sex with men. In order to do this, we used data from UK Biobank, which is a research database that contains in-depth health information from approximately half a million people in the UK. We found no differences in the number of new cases for breast, prostate and colorectal cancer, but we did find that sexual minority adults have a higher risk for lung cancer, due to greater exposure to smoking.

**Abstract:**

Background. There is limited evidence about cancer incidence for lesbian, gay and bisexual women and men, although the prevalence of cancer risk factors may be higher. Aim. To describe cancer incidence for four common cancers (breast, lung, colorectal and prostate). Methods. This project used UK Biobank participant data. We explored risk factor prevalence (age, deprivation, ethnicity, smoking, alcohol intake, obesity, parity, and sexual history), and calculated cancer risk, for six groups defined based on sexual history; women who have sex exclusively with men (WSEM), or women (WSEW), women who have sex with men and women (WSWM); men who have sex exclusively with women (MSEW), or men (MSEM), and men who have sex with women and men (MSWM). Results. WSEW, WSWM, MSEM, and MSMW were younger, more likely to smoke, and to live in more deprived neighbourhoods. We found no evidence of an association between sexual history and breast, colorectal, or prostate cancer in age-adjusted models. Lung cancer incidence was higher for WSWM compared with WSEM, HR (95%CI) 1.78 (1.28–2.48), *p* = 0.0005, and MSWM compared with MSEW, 1.43 (1.03–1.99), *p* = 0.031; after adjustment for smoking, this difference was no longer significant. Conclusions. Sexual minority groups have a higher risk for lung cancer, due to greater exposure to smoking.

## 1. Introduction

There is limited evidence about cancer and other health outcomes for lesbian, gay and bisexual (LGB) people in the UK [1]. What research there is suggests poorer outcomes than for people who are heterosexual [2,3,4]. Much LGB health research to date has had a focus on sexually transmitted infections (STI), especially HIV/AIDs, and evidence on health and health care outcomes for long-term conditions including cancer is still emerging.

Working with the public when developing research improves both the quality and relevance of work, particularly when it relates to minority groups [5,6]. It enables a focus on issues important to the lives of patients and allows patients to offer their relevant insights of specific health conditions or of belonging to a specific group. For this project, during 2020–2022, we worked with an LGBTQ+ research involvement group to identify research priorities. The group identified cancer, and particularly lung, breast, and prostate cancer, as cancers they were interested in, or concerned about [7,8,9]. Along with colorectal cancer, these are the four commonest cancer diagnoses in the UK. The group also highlighted higher rates of smoking among sexual minority adults, and its link to cancer, as a key area of concern. Smoking is an important risk factor for cancer [10], and rates of smoking are higher amongst sexual minority groups, particularly LGB women [11,12].

Regarding cancer outcomes, cross-sectional studies exploring overall cancer prevalence among LGB adults found variation by sex, with higher cancer prevalence among gay men [13,14], and site-specific cancer research has highlighted disparities for rarer cancers that are causally linked to HIV [15,16] and human papilloma virus transmission (HPV), e.g., cervical, anal, and oropharyngeal cancer [16,17,18], rather than cancer diagnoses with most frequent incidence.

Poorer health and cancer outcomes experienced by LGB adults have multifactorial causes, both structural and individual [19]. LGB adults are more likely to live in areas of higher socio-economic deprivation [2]. Minority stress [20], where social prejudices, stigma, and discrimination contribute to an environment where health is adversely impacted is a second important dimension. Higher levels of risk factors for common cancers, developed in part through the complex interplay with minority stress, including obesity, smoking and high alcohol intake [21], are a third dimension. Parity, an important protective factor for breast cancer [22], is lower in sexual minority women [23].

Disparities in access to and quality of health care offered to sexual minority adults compared with their heterosexual counterparts are another contributor [2,24,25].

Although data, particularly cross-sectional survey data collections, are improving, and the UK National Cancer Registration and Analysis Service added sexual orientation to its core demographic dataset in 2018, this field is not yet mandatory, and longitudinal studies with an adequate sample size and collection of sexual identity as well as health and cancer outcome data are rare [26,27,28,29].

UK Biobank is a longitudinal cohort study which, between 2006 and 2010, collected rich information about cancer risk factors, and about sexual history, from 500,000 40–69 year olds [30]. Consent for data linkage was also collected and cancer diagnoses are linked from national cancer registration data, making this a valuable resource for understanding cancer risk among older sexual minority adults.

In order to address the lack of data on cancer outcomes from LGB adults, and building on patient and public identified priorities, as well as an understanding of known disparities in risk factors, this analysis used UK Biobank to answer the specific research question, ‘Are there differences in breast, lung, colorectal and prostate cancer incidence between sexual minority and heterosexual patients, and are such differences attributable to known risk factors?’.

## 2. Materials and Methods

### 2.1. Data

We evaluated cancer incidence among women who have sex with women and men who have sex with men using data from UK Biobank. At the time of recruitment to UK Biobank, participants were resident within 25 miles of 1 of 22 assessment centres in England (89% of participants), Scotland (7%), or Wales (4%) [31]. Data on a range of lifestyle factors were collected through a touch-screen questionnaire at baseline, and on cancer outcomes through linkage to national cancer registries. Full information on the study have been reported elsewhere [31,32]. The data used for this study were accessed through UK Biobank (application ID 42861).

### 2.2. Cancer

Using linked cancer registry data [33], breast cancer cases were defined with an ICD10 code of C50 or an ICD9 code of 1740, 1750, 1741, 1742, 1743, 1744, 1745, 1746, 1748, 1749, or 1759. Prostate cancer was defined with an ICD10 code of C61 or an ICD9 code of 185. Colon cancer was defined with an ICD10 code of C18 or an ICD9 code of 1530, 1531, 1532, 1533, 1534, 1535, 1536, 1537, 1538, or 1539 and rectal cancer with ICD10 codes of C19 or C20, or an ICD9 code of 1540 or 1541; for analysis, these two cancers were combined to consider all colorectal cancer together. Lung cancer was defined by an ICD10 diagnosis code of C33 or C34 or an ICD9 code of 1620, 1622, 1623, 1624, 1625, 1628, or 1629. Both prevalent and incident cases were identified. Prevalent cases of all cancers were defined as any diagnosis with a cancer-registry-recorded date of diagnosis that occurred before the date of baseline assessment for the UK Biobank cohort (study entry), and people with prevalent cancer were excluded in the analysis of incident cancer (first recorded cancer diagnosis after study entry) on a cancer-by-cancer basis.

### 2.3. Sexual History

Sexual identity (whether someone identifies as lesbian, gay, or bisexual) overlaps with an individual’s sexual history [34]; at baseline assessment, UK Biobank only collected information about sexual history, and not on any other measure of sexual orientation. In previous work, we evaluated the use of sexual history for understanding health inequalities experienced by sexual minority women and men, and found that the measure, although not perfect, was adequate [30].

In detail, during the baseline touch-screen questionnaire, participants answered questions about their same and opposite-sex sexual history and were given the option to not answer these questions if they found them too sensitive. Participants were asked their total number of sexual partners, ‘About how many sexual partners have you had in your lifetime?’ and about their history of same-sex sex, ‘Have you ever had sexual intercourse with someone of the same sex?’ Participants indicating a history of same-sex sex were asked their number of same-sex sexual partners, ‘How many sexual partners of the same sex have you had in your lifetime?’ These responses were used to identify women and men who had never had sex, women who have sex exclusively with men (WSEM), or women (WSEW), women who have sex with men and women (WSWM); men who have sex exclusively with women (MSEW), or men (MSEM), and men who have sex with women and men (MSWM) (details in Appendix A). Women and men who had never had sex, or with missing responses to sexual history question were excluded from all analyses.

### 2.4. Cancer Risk Factors

The following known socio-demographic and health-related cancer risk factors were considered; age at baseline assessment, sex, smoking, alcohol consumption, ethnicity, deprivation, body mass index (BMI), parity in women, lifetime number of sexual partners and age at first intercourse.

This analysis used the ‘sex’ field rather than ‘genetic’ sex to classify participants as women or men. The sex field is acquired from central registry at recruitment, but in some cases updated by the participant. [35] Ethnicity was categorised based on five groups defined by the UK Office of National Statistics, and deprivation from the Townsend score for each participant, a small area-based measure of socio-economic deprivation, categorised using national quintile-defining cut-off points.

Smoking status was derived from responses to two questions ‘Do you smoke tobacco now?’, with responses ‘Yes, on most or all days’, ‘Only occasionally’, ‘No’ and ‘Prefer not to answer’; and ‘In the past, how often have you smoked tobacco?’, with responses ‘Smoked on most or all days’, ‘Smoked occasionally’, ‘Just tried once or twice’, ‘I have never smoked’ and ‘Prefer not to answer’. Informative responses were used to classify participants as ‘Current’, ‘Previous’, or ‘Never’ smokers.

Participants were asked about alcohol consumption, ‘About how often do you drink alcohol?’: ‘Daily or almost daily’, ‘Three or four times a week’, ‘Once or twice a week’, ‘One to three times a month’, ‘Special occasions only’, ‘Never’ and ‘Prefer not to answer’. ‘One to three times a month’ and ‘Special occasions only’ responses were combined in this analysis.

Continuous BMI was calculated using nurse-measured height and weight, then categorised with descriptors according to the WHO guidelines: <20 (underweight or lower healthy weight, aggregated because of low numbers), 20–24.9 (healthy weight), 25–29.9 (overweight), and 30+ (obese) [36].

Women were asked ‘How many children have you given birth to?’ and ‘Have you ever had any stillbirths, spontaneous miscarriages or terminations?’. Parity was calculated as the number of live-born children plus number of stillbirths and categorised into the following groups: parity 0; parity 1; parity 2; and parity 3+. Age of first sexual intercourse and the lifetime number of sexual partners were recorded from the initial sexual history measures.

### 2.5. Analysis

#### 2.5.1. Descriptive Statistics

In our first analysis, we described cancer risk factor prevalence stratified by sex and sexual history for all participants where sexual history could be defined. Missing data for each cancer risk factor were excluded on a risk factor-by-risk factor basis.

#### 2.5.2. Cancer Incidence

Prevalent cancer cases (on a cancer-by-cancer basis) and individuals with missing cancer risk factor data for any risk factor (except age of first intercourse, which was considered only in sensitivity analysis because of the high amount of missing data) were excluded for all unadjusted and adjusted analyses of cancer incidence.

Unadjusted breast, lung, colorectal and prostate cancer incidence were estimated stratified by sex and sexual history.

Person-time at risk for cancer incidence was calculated as time (in years) between baseline assessment, and the earliest of, the date of first site-specific incident cancer diagnosis (cases), date of death, or the date of last cancer registry linkage (censoring). The date of last cancer registry linkage is 29 February 2020 in England and Wales, and 31 January 2021 in Scotland, and we used the earlier date of 29 February 2020 in this analysis.

#### 2.5.3. Multivariable Analysis

For each cancer, we used Cox regression and considered three models, unadjusted, adjusted for age, and fully adjusted for all significant cancer risk factors to explore the relationship between sexual history and common cancer incidence in UK Biobank for each cancer diagnosis, stratified by sex. The fully adjusted model, separately for women and men, for each diagnosis, was developed using a backward Akaike Information Criteria (AIC) stepwise function. Beginning with a ‘maximum’ model including all potential covariates (age, ethnicity, smoking, alcohol, parity, deprivation, BMI, and lifetime number of sexual partners), the AIC function simplifies the model by removing variables that do not significantly improve the model fit. The function was not permitted to exclude the sexual history variable regardless of its effect on model fit. All covariates were included as categorical variables in the multivariable analysis, except for age, where we included it as a continuous measure (with quadratic terms where this improved model fit).

#### 2.5.4. Sensitivity Analyses

We explored the impact of missing data on these analyses in a series of sensitivity analyses.

All analyses were carried out using R version 4.2.1.

## 3. Results

### 3.1. Descriptive Statistics

After excluding respondents with missing sexual history, 403,637 UK Biobank participants were included in descriptive analysis including 214,801 WSEM, 700 WSEW and 5161 WSWM, 176,591 MSEW, 2111 MSEM, and 4273 MSWM. Analyses of cancer incidence were carried out on the smaller sample of 397,681 responses for whom complete covariate information was available (210,464 WSEM, 5048 WSWM, 688 WSEW, 175,162 MSEW, 2090 MSWM, 4229 MSEM), details Appendix A
Figure A1.

MSEW and WSEM are generally older than other groups with the median (IQR) age in WSEM 57 (50–63), WSEW 50 (45–58), WSWM 51 (45–57), MSEW 58 (50–63), MSEM 51 (45–60), and MSWM 53 (47–61), and live in less deprived areas, with the median deprivation quintile in the study cohort the second least deprived for WSEM, WSEW and MSEW, the middle group for WSWM and MSEM, and the second most deprived for MSWM (Table 1, full details of counts in each group in Appendix B, Table A1). Men and women with a history of both same and opposite-sex sex are more likely to report daily or almost daily alcohol intake than those with a history of only same or opposite-sex sex, and to be current or previous smokers. WSEW (16.7%) and WSWM (21.1%) are much more likely be current or previous smokers than WSEM (11.5%), with similar patterns among men. WSEW (68.4%) and WSWM (43.4%) are much more likely than WSEM (15.6%) to have never had children.

### 3.2. Unadjusted Incidence

The number of incident cases and incidence rates are summarised in Table 2 and Table 3. Median length of follow up in all groups was 11 years (IQR 10–12). Breast cancer incidence in all women was higher than lung or colorectal, and in men prostate cancer incidence was highest. Cancer incidence for all diagnoses was highest in WSEM and MSEW, reflecting the older age profile of these groups (Table 1).

### 3.3. Multivariable Analysis

#### 3.3.1. Women

There was no evidence of an association between sexual history and breast cancer risk (*p* > 0.08 in all models); although the incidence for WSEW breast cancer was lower in all models, this was not significant (Table 4).

In an analysis adjusting only for age, we found higher lung cancer incidence among women with a history of both same and opposite-sex sex (*p* = 0.0022) HR (95%CI) 1.77 (1.28–2.46) and women with a history of same-sex sex only 2.06 (0.92–4.60). This was explained by a higher cancer smoking prevalence in fully adjusted models (Table 4 and Appendix C, Table A2). There was no evidence of an association between sexual history and incident colorectal cancer risk (*p* > 0.2 in all models).

#### 3.3.2. Men

After accounting for differences in age, we found no evidence of an association between prostate, lung or colorectal cancer incidence with sexual history (Table 5). For MSWM, there was evidence 1.42 (1.02–1.97), *p* = 0.031 (stratum-specific *p*-value), that lung cancer incidence was higher than for MSEW, which disappeared after adjusting for smoking history (Table 4, Appendix C). Adjusting for cancer risk factors had a large impact on the size of the estimates for lung cancer; and for MSWM, this explained the higher cancer incidence seen in this group. Although prostate cancer rates are lower among men who have sex with men, this is explained by the lower age of this group.

Results were consistent in sensitivity analyses exploring the impact of missing data (Appendix D and Appendix E, Table A3, Table A4, Table A5 and Table A6).

## 4. Discussion

We explored cancer incidence for four common cancers among women who have sex with women and men who have sex with men using UK Biobank with linked cancer incidence data from national (UK) cancer registries. We found no evidence of an association between sexual history and breast, colorectal, or prostate cancer incidence in age-adjusted models. Lung cancer incidence was higher for WSWM compared with WSEM, and MSWM compared with MSEW; accounting for differences in smoking prevalence in these groups explained this higher incidence.

This study builds on existing knowledge about the higher prevalence of cancer risk factors, particularly smoking, among lesbian, gay and particularly bisexual adults [28,37,38,39,40,41], and adds new evidence about how these known disparities translate to poorer health outcomes.

Of the common cancer diagnoses we considered—lung, colorectal, breast, and prostate—we found disparities in age-adjusted incidence only for lung cancer. This provides additional evidence which builds on our earlier study which found that for most common and rarer cancer sites (except for those associated with HPV or HIV), the odds of specific cancer site diagnosis seemed to be independent of sexual orientation [16]. The novel finding here for lung cancer probably reflects that this analysis includes controls without cancer, rather than a case-only framework used in earlier work, which may have underestimated disparities associated with smoking as all study participants had a cancer diagnosis. However, it is not surprising that of all the cancers considered it is lung cancer where disparities in cancer risk factors among sexual minority women and men translate into poorer health outcomes. Of all common cancers, lung cancer is the one with the largest attributable fraction due to known lifestyle and environmental risk factors (chiefly smoking) [42,43].

Previous systematic reviews have highlighted the lack of high-quality data, and the importance of prospective cohort studies in producing evidence to better understanding breast cancer risk in lesbian and bisexual women [44] This study goes part of the way towards addressing this evidential need. Although we find no evidence of an association between sexual history and breast cancer incidence at this time, we do note that the sample size of included WSEW and WSWM was relatively small; further analysis as the UK Biobank cohort continues to be followed up over time may be warranted.

Despite our finding of no evidence of disparities in prostate cancer incidence for MSM, we do note that previous research has found that quality of life and some outcomes after prostate cancer treatment are poorer in this group; future research could explore whether lifestyle differences identified in this analysis have impact on cancer outcomes beyond cancer incidence considered in this research [45].

This research has important implications for both health policymakers and in clinical practice. In clinical practice, there is evidence that lesbian, gay and bisexual adults have poorer experiences of cancer care [25]; the evidence from this work that lesbian, gay and bisexual women and men are likely to be over-represented among people with lung cancer highlights the importance of work to improve holistic cancer care for LGB communities [46]. From a public health practice, and policy perspective, our work highlights that approaches to support smoking cessation among LGB communities will be particularly important and have the potential to make material differences to the inequalities in health outcomes experienced by sexual minority adults. It is also important to highlight that the higher prevalence of smoking among LGB adults arises from a complex interplay of minority stress and experiences of stigma and discrimination [19], and the historic targeting of LGBT communities by tobacco companies [47]; smoking cessation support is a practical area for redressing these inequalities.

There are limitations to this analysis; we describe inequalities experienced by women who have sex with women, and men who have sex with men, and sexual orientation was not recorded directly; previous research has found this is a reasonable approach [30], but future data collections should consider additionally including sexual orientation (and gender identity) as a priority. Having said this, it is important to acknowledge that sexual orientation alone is not inherently a risk factor for cancer, instead it is a proxy measure for the impact of experiencing chronic, toxic stress as well as health diminishing coping behaviours used to manage the negative experiences caused by heterosexism, and multi-level discrimination, stigma, and marginalisation.

Additionally we focus only on the four commonest cancers; disparities may exist for other cancers, for example, liver and oesophageal cancers, which are associated with alcohol consumption, which we find in this analysis, consistent with previous research [28], to be at higher levels among women who have sex with women and men who have sex with men.

In an earlier analysis, we explored the characteristics of UK Biobank participants with missing information about sexual history [30]. People with missing sexual history were older, and less likely to live in the least deprived parts of the UK, compared with women and men who have an exclusively opposite-sex sexual history. Although it is a limitation that approximately 20% of UK Biobank participants with missing sexual history were excluded in these analyses, we did adjust for both age and deprivation in multivariable analysis, mitigating this impact on the study findings. We were not able to explore disparities in cancer outcomes for trans and gender-diverse adults, and this is a further limitation. A strength of this research, however, is that it built on strong patient and public involvement [7,8,9], which is important to ensure that there is not a mismatch between research and the priorities of the communities it is designed to support [48].

## 5. Conclusions

Women who have sex with women and men who have sex with men are higher-risk groups for lung cancer due to greater exposure to smoking.

## Figures and Tables

**Table 1 cancers-15-02031-t001:** Cohort characteristics, stratified by sexual history (%).

	Women Who Have Sex Exclusively with Men (n = 214,801)	Women Who Have Sex Exclusively with Women (n = 700)	Women Who Have Sex with Women and Men (n = 5161)	Men who Have Sex Exclusively with Women (n = 176,591)	Men Who Have Sex Exclusively with Men (n = 2111)	Men Who Have Sex with Women and Men (n = 4273)
**Age (n = 403,637)**						
35–44	10.3	22.4	21.9	10.6	21.6	16.6
45–49	13.6	25.9	23.7	12.9	21.5	18.1
50–54	16.0	17.7	20.6	14.7	16.9	17.6
55–59	18.9	13.4	16.5	17.8	14.1	17.2
60–64	24.3	13.0	12.6	24.3	14.7	18.1
65+	16.9	7.57	4.71	19.8	11.3	12.3
**Ethnicity (n = 402,166)**						
White	96.0	91.7	95.1	96.0	94.2	93.2
Mixed	0.48	1.15	1.52	0.33	0.76	1.18
Asian	1.18	2.58	0.49	1.75	2.05	2.57
Black	1.33	1.72	1.76	1.05	0.95	1.41
Other/Chinese	1.00	2.87	1.17	0.82	2.05	1.65
**Deprivation (n = 403,127)**						
Least Deprived	38.9	26.5	21.3	39.7	16.5	21.2
2	21.4	20.2	16.0	21.2	12.5	16.6
3	14.8	16.5	18.0	14.5	15.2	13.6
4	12.9	15.3	20.6	12.5	20.9	20.5
Most Deprived	12.0	21.6	24.1	12.1	34.9	28.1
**Smoking (n = 402,633)**						
Current	8.29	14.6	19.2	11.5	16.7	21.1
Previous	31.1	30.8	43.7	38.8	31.3	40.7
Never	60.6	54.6	37.2	49.6	51.9	38.2
**Alcohol (n = 403,480)**						
Daily or almost daily	16.5	16.9	23.3	26.3	26.9	30.6
3 or 4 times a week	21.4	19.4	22.9	27.1	22.5	22.2
1 or 2 times a week	26.1	24.4	23.4	25.7	23.0	22.4
1 to 3 times a month	27.5	27.0	23.9	15.5	20.7	17.4
Never	8.43	12.3	6.53	5.39	6.87	7.31
**BMI (n = 403,637)**						
Under 20	3.32	2.86	3.89	0.89	2.37	1.47
20–25	36.6	33.6	37.7	24.2	33.1	27.7
25–30	36.6	36.3	34.1	49.6	42.2	45.8
30+	23.5	27.3	24.3	25.4	22.4	25.0
**Lifetime Sexual Partners (n = 403,637)**						
1	33.8	25.9	0.775	24.5	9.52	1.40
2 to 3	26.2	25.4	12.6	22.0	11.7	11.3
4 to 5	16.7	19.6	15.4	15.8	10.8	13.8
6+	23.2	29.1	71.2	37.7	67.9	73.5
**Age of First Intercourse (n = 396,622)**						
<15	2.51	3.42	10.2	4.92	11.9	15.1
15–17	31.3	24.1	41.9	30.0	20.0	32.3
18+	66.2	72.5	47.9	65.1	68.2	52.6
**Parity (n = 400,745)**						
0	15.6	68.4	43.4	100	100	100
1	12.9	5.19	15.5	0	0	0
2	45.4	16.9	26.9	0	0	0
3+	26.1	9.52	14.2	0	0	0

**Table 2 cancers-15-02031-t002:** Incident cancer case numbers, and incidence rates in women, stratified by sexual history.

	Cases	Overall Cancer Incidence in UKBB per 1000 Person-Years Follow up
**Breast**		
Women who have sex exclusively with men	6921	3.10 (3.03–3.17)
Women who have sex with women and men	158	2.92 (2.48–3.41)
Women who have sex exclusively with women	14	1.91 (1.04–3.20)
**Lung**		
Women who have sex exclusively with men	1444	0.616 (0.585–0.649)
Women who have sex with women and men	38	0.676 (0.478–0.928)
Women who have sex exclusively with women	6	0.790 (0.290–1.72)
**Colorectal**		
Women who have sex exclusively with men	1892	0.811 (0.775–0.849)
Women who have sex with women and men	34	0.606 (0.291–1.73)
Women who have sex exclusively with women	6	0.794 (0.420–0.846

**Table 3 cancers-15-02031-t003:** Incident cancer case numbers, and incidence rates in men, stratified by sexual history.

	Cases	Overall Cancer Incidence in UKBB per 1000 Person-Years Follow up
**Prostate**		
Men who have sex exclusively with women	8142	4.42 (4.33–4.52)
Men who have sex with women and men	166	3.74 (3.19–4.35)
Men who have sex exclusively with men	72	3.28 (2.57–4.13)
**Lung**		
Men who have sex exclusively with women	1423	0.748 (0.709–0.788)
Men who have sex with women and men	39	0.855 (0.609–1.17)
Men who have sex exclusively with men	15	0.664 (0.372–1.10)
**Colorectal**		
Men who have sex exclusively with women	2427	1.29 (1.24–1.34)
Men who have sex with women and men	50	1.10 (0.819–1.45)
Men who have sex exclusively with men	21	0.936 (0.580–1.43)

**Table 4 cancers-15-02031-t004:** Models of breast, lung, and colorectal cancer in women.

	Unadjusted	Age Adjusted *	Full Model **
	Hazard Ratio (95% CI)	Hazard Ratio (95% CI)	Hazard Ratio (95% CI)
**Breast Cancer**			
WSEM	1 (reference)	1 (reference)	1 (reference)
WSWM	0.91 (0.77–1.07)	1.01 (0.86–1.19)	0.93 (0.79–1.10)
WSEW	0.63 (0.37–1.06)	0.70 (0.41–1.18)	0.63 (0.37–1.07)
Joint *p*-value	0.087	0.35	0.13
**Lung Cancer**			
WSEM	1 (reference)	1 (reference)	1 (reference)
WSWM	1.09 (0.79–1.51)	1.77 (1.28–2.46)	1.15 (0.82–1.59)
WSEW	1.30 (0.59–2.91)	2.06 (0.92–4.60)	1.56 (0.70–3.48)
Joint *p*-value	0.73	0.0022	0.44
**Colorectal Cancer**			
WSEM	1 (reference)	1 (reference)	1 (reference)
WSWM	0.75 (0.53–1.06)	1.06 (0.75–1.50)	1.10 (0.78–1.56)
WSEW	1.01 (0.45–2.24)	1.40 (0.63–3.12)	1.37 (0.61–3.06)
Joint *p*-value	0.23	0.70	0.67

* Breast and lung cancer models adjusted with linear and quadratic age term. Colorectal cancer model adjusted with linear age term only. ** Full breast cancer model adjusted for age at recruitment (linear and quadratic terms), ethnicity, smoking, alcohol, parity, BMI. Lung cancer model adjusted for age at recruitment (linear and quadratic terms), Townsend category, smoking, alcohol, and BMI. Colorectal cancer model adjusted for age at recruitment (linear term only), smoking, lifetime number of sexual partners, BMI.

**Table 5 cancers-15-02031-t005:** Models of prostate, lung, and colorectal cancer in men.

	Unadjusted	Age Adjusted *	Full Model **
	Hazard Ratio (95% CI)	Hazard Ratio (95% CI)	Hazard Ratio (95% CI)
**Prostate Cancer**			
MSEW	1 (reference)	1 (reference)	1 (reference)
MSWM	0.84 (0.72–0.98)	1.07 (0.92–1.25)	1.10 (0.94–1.28)
MSEM	0.75 (0.59–0.94)	1.09 (0.86–1.37)	1.10 (0.87–1.39)
Joint *p*-value	0.0027	0.54	0.39
**Lung Cancer**			
MSEW	1 (reference)	1 (reference)	1 (reference)
MSWM	1.10 (0.79–1.52)	1.42 (1.02–1.97)	0.90 (0.64–1.25)
MSEM	0.90 (0.54–1.49)	1.31 (0.79–2.18)	1.03 (0.62–1.71)
Joint *p*-value	0.79	0.087	0.79
**Colorectal Cancer**			
MSEW	1 (reference)	1 (reference)	1 (reference)
MSWM	0.85 (0.64–1.13)	1.05 (0.79–1.40)	1.03 (0.77–1.36)
MSEM	0.74 (0.48–1.14)	1.01 (0.66–1.56)	1.05 (0.68–1.61)
Joint *p*-value	0.18	0.94	0.96

* Prostate, lung, and colorectal cancer models adjusted with linear and quadratic age term. ** Full prostate cancer model adjusted for age at recruitment (linear and quadratic term), ethnicity, smoking, alcohol, and BMI. Lung cancer model adjusted for age at recruitment (linear and quadratic term), Townsend category, smoking, lifetime number of sexual partners, alcohol, and BMI. Colorectal cancer model adjusted for age at recruitment (linear term and quadratic term), ethnicity, smoking, alcohol, and BMI.

## Data Availability

This research was a secondary data analysis of data from UK Biobank, as specified in the Acknowledgements Section.

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
