# Peer review of "Breast, Prostate, Colorectal, and Lung Cancer Incidence and Risk Factors in Women Who Have Sex with Women and Men Who Have Sex with Men: A Cross-Sectional and Longitudinal Analysis Using UK Biobank"

_cancers, 2023, doi:10.3390/cancers15072031_

Round 1

Reviewer 1 Report

This is a well-conceptualized paper examining rates of four common cancers among individuals of differing sexual behaviors in the UK Biobank (N=500,000).  I have a few minor suggestions:

- Was any attempt made to compare the ~93,000 people who declined to answer the sexual behavior question to the rest of the population in terms of cancer risk and incidence?  This may be a somewhat unique group.

- Some additional references may help to contextualize the history of epidemiologic studies of cancer among sexual minority populations, including Frisch et al., 2003, "Cancer in a Population-based Cohort of Men and Women in Registered Homosexual Partnerships," Boehmer et al., 2011, "Cancer survivorship and sexual orientation," and others.

- A minor point: there is an inconsistent use of both commas and hyphens, e.g., "orien-tation" line 267.

Otherwise the article provides important information and would be of interest to readers.

Author Response

Please see the attached response document which includes our responses to all three sets of (very helpful) reviewer comments

Reviewer 2 Report

Outcome for PCa MSM patients are demonstrably worse. Highlighting this might encourage researchers to look at minority stress and life style differences in future studies. 

Author Response

(The authors gave the same response as above.)

Reviewer 3 Report

Thank you for the opportunity to review the manuscript entitled, “Breast, prostate, colorectal, and lung cancer incidence and risk factors in sexual minority women and men: a cross sectional and longitudinal analysis using UK biobank” for consideration for publication in the journal Cancers. This manuscript reports on the incidence of four cancers that LGB community identified as high priority cancer sites including colorectal cancer, prostate cancer, lung cancer, and breast cancer.  Results indicate that there were no differences in cancer incidence for any cancer sites except lung cancer. Individuals identified as WSWM and MSWM each had higher rates of lung cancer than other groups. Further, this disparity was mediated when smoking status was added to the multivariable model, indicating that smoking explained the lung cancer disparity.  There are a number of limitations and concerns that must be addressed before this manuscript can be fully considered for possible publication. These minor and major concerns are listed below. Thank you.

Minor Concerns:

·         In the limitations section, please address the methodological and theoretical limitations of measuring sexual orientation with sexual behavior questions that exclude identity and attraction.

·         Methods: please revise to include how measures for smoking and alcohol use were scored and what the meaning of scores were (ie., averaged (or summed or something else) and high score was higher risk) and please include any reverse coding that was required.

Major Concerns:

·         Abstract

o   the second sentence in the abstract (lines 13-14) names that we know more about causal linkages between cancer and HIV and HPV, than with ‘common’ cancers among LGBTQ+ adults. One note, neither HIV nor HPV are cancers, but they are causally associated with cancers, including colorectal cancer (HPV). So, please revise this to be more specific and accurate about risk factors and actual cancer.

o   If using “LGB” to refer to the target group throughout the manuscript, please revise title for consistency. I think that LGB is far more specific and less marginalizing language than ‘sexual minority’.

·         Introduction

o   Please write in a focused way. In line 34, please revise to exclude “other health outcomes” if this paper is meant to be specifically about cancer and cancer risk behaviors.

o   Line 40: instead of ‘common cancer diagnoses’ I recommend using language that is supported by annual cancer reports about most frequent cancer incidence. The word ‘common’ lacks the necessary and expected scientific specificity.

o   Lines 41-54: There is a LOT of dense, important information here. But it moves too quickly and is too underdeveloped for readers to follow and understand how the multiple factors contribute to risk. Slow down the language by supporting each point. This may require a revision that involves turning this paragraph into multiple paragraphs. I recommend thinking about this in terms of a framework with multiple levels. Also, Fredricksen-Goldsen’s et al. health promotion and equity model may be a useful starting place for organizing these ideas.

§  Starting with line 50: this sentence that describes the data source should be an entirely new paragraph.

o   Unclear who made up the public involvement group, how they were selected, when they came together, how they arrived at consensus etc.  This paragraph (lines 55-64) come out of left field. Somehow the structure and approach used via this method needs to come earlier and shape/structure the introduction. It is possible that the advisory group piece may need to go in methods since they contributed to identifying the cancers of high incidence. Revise for clarity.

·         Results

o   Table 1: please include the full text description/name of each column in Table 1. For individuals who do not regularly work with these acronyms, reading this table and switching back to the introduction to recall which group is being reported on, will be frustrating and confusing. This revision will improve readability for a wider readership.

o   Table 2: please revise to include full text description for each row.

·         Discussion:

o   Please revise the Discussion section to include evidence that shows bisexual identified people have both higher in the forms of smoking and alcohol use compared to lesbian and gay identified people and that other publications have found that bisexual identified people have higher incidence for cancer.

§  Quinn, GP., et al. (2015). Cancer and lesbian, gay, bisexual, transgender LGBT populations, CA: Cancer Journal for Clinicians, 65(5).

§  Margolies, L., and Brown, C.G. (2018). Current state of knowledge about cancer in LGBT people, Seminars in Oncology Nursing, 34(1), pp 3 – 11.

·         Please revise the Discussion to include acknowledgment that sexual orientation is not inherently a risk factor for cancer or any disease. Rather, sexual orientation is often used as a proxy for the potentially deleterious effects of experiencing chronic, toxic stress as well as possible health diminishing coping behaviors used to manage the experience of negative and toxic stress caused by heterosexism and multi-level discrimination, stigma, marginalization etc.

Author Response

(The authors gave the same response as above.)

Round 2

Reviewer 3 Report

No additional comments for revision. Thank you for taking previous comments into consideration and for the high quality revision.